# Visual Manipulation with Legs

**Xialin He**[†,1], **Chengjing Yuan**[†,2], **Wenxuan Zhou**[3], **Ruihan Yang**[2],
**David Held**[3], **Xiaolong Wang**[2]

[1]University of Illinois Urbana-Champaign    [2]UC San Diego    [3]Carnegie Mellon University
[†]Equal Contributions
Page: Visual-Manipulation-With-Legs

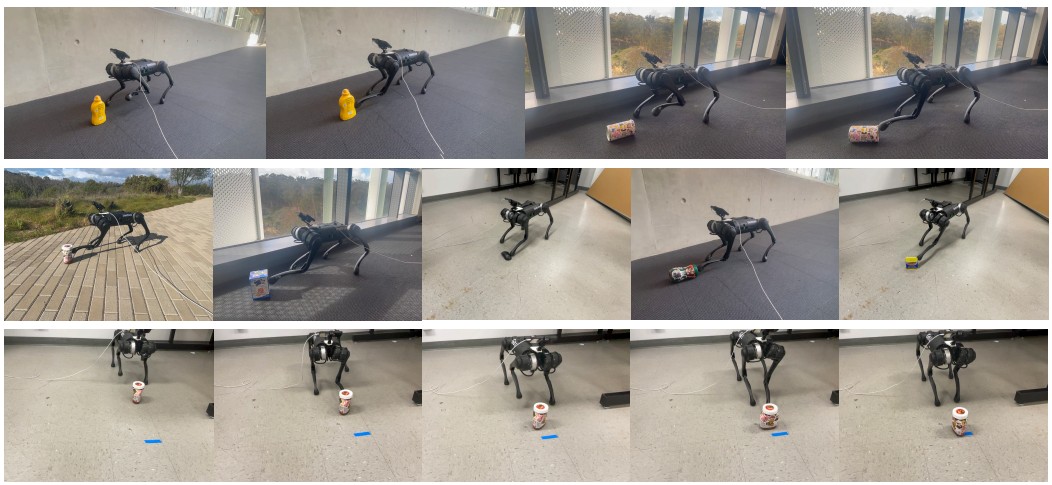

Figure 1: We propose a system that enables legged robots to interact with various objects, moving them to distant goals through repeated pushing and walking.

**Abstract:** Animals use limbs for both locomotion and manipulation. We aim to equip quadruped robots with similar versatility. This work introduces a system that enables quadruped robots to interact with objects using their legs, inspired by non-prehensile manipulation. The system has two main components: a visual manipulation policy module and a loco-manipulator module. The visual manipulation policy, trained with reinforcement learning (RL) using point cloud observations and object-centric actions, decides how the leg should interact with the object. The loco-manipulator controller manages leg movements and body pose adjustments, based on impedance control and Model Predictive Control (MPC). Besides manipulating objects with a single leg, the system can select from the left or right leg based on critic maps and move objects to distant goals through base adjustment. Experiments evaluate the system on object pose alignment tasks in both simulation and the real world, demonstrating more versatile object manipulation skills with legs than previous work. Videos can be found on project website.

**Keywords:** Reinforcement Learning, Loco-Manipulation, Legged Robot

## 1   Introduction

In robotics, manipulation is often associated with robot arms and locomotion with legs. Typically, quadruped robots use dedicated arms for manipulation, while legs are used for walking [1, 2, 3]. However, animals seamlessly use their limbs for both tasks. Primates adeptly utilize all four limbs for walking, climbing, and object manipulation, suggesting quadruped robots could leverage their legs for a broader range of tasks, simplifying design and expanding manipulation capabilities.

This work bridges the gap between locomotion and manipulation by leveraging non-prehensile manipulation for leg-based tasks. Non-prehensile manipulation, which involves tasks without grasping,

8th Conference on Robot Learning (CoRL 2024), Munich, Germany.

aligns with the capabilities of quadruped robot legs [4]. Unlike previous efforts focused on simpler tasks like kicking a ball or pushing a button [5, 6, 7, 8, 9], our task demands more precise control.

Our system for solving object manipulation tasks with legs comprises a learned visual manipulation policy and a model-based loco-manipulation controller. The manipulation module, trained with reinforcement learning in simulation using point cloud observations, dictates how the leg interacts with the object to achieve the goal pose. We utilize an object-centric action space based on point cloud data [10], allowing precise leg manipulation. The policy determines a contact point and motion vector, and critic values assist in choosing between the left or right leg.

The model-based loco-manipulation controller synchronizes leg movements and body adjustments to execute commands. It translates high-level actions from the visual manipulation policy into low-level torque commands. The manipulation leg uses impedance control, while other legs maintain stability with Model Predictive Control (MPC). The controller adjusts the base pose when the object is out of reach, facilitating effective pushing toward distant goals.

We evaluate the system on various object pose alignment tasks, demonstrating the visual manipulation policy's effectiveness against baselines, its generalization to unseen objects, and the advantages of leg selection. Real-world experiments assess the feasibility of zero-shot sim2real transfer and the system's precision in moving objects toward distant goals, showing improvements over previous work [5, 6, 7, 8, 9].

## 2 Related Work

**Quadrupedal Locomotion:** Legged robots have traditionally relied on proprioceptive senses for navigating complex terrains, using model-based methods that ensure stability but require significant computational power [11, 12, 13, 14, 15, 16, 17, 18]. Recent advancements in model-free reinforcement learning have reduced computational demands while enhancing generalization [19, 20, 21, 22, 23, 24, 25]. Although these techniques have primarily been applied to locomotion, recent work has integrated exteroceptive perception, such as elevation maps and depth sensing, to enhance high-level planning [26, 27, 28, 29, 30, 31, 32, 33, 34]. While most research uses these approaches for locomotion, our work uniquely applies them to dexterous manipulation, leveraging model-based controllers for precise end-effector positional control.

**6D Manipulation with Manipulators:** Previous work in 6D manipulation has focused on dexterous hands and grippers on fixed bases designed for manipulation [35, 36]. In contrast, we use the leg of a robot dog for 6D manipulation, an underexplored approach.

**Non-Prehensile Manipulation:** Non-prehensile manipulation, which involves interacting with objects without grasping, has been a focus for tasks like pushing, sliding, and flipping. Foundational work by Mason [4] established the mechanics of these tasks, while more recent approaches have integrated reinforcement learning and model-based methods to enhance precision and adaptability [37, 38]. For instance, Zhou et al. [39] developed learning frameworks for robust manipulation in unstructured environments. Building on these advances, our work applies non-prehensile techniques in legged robots, achieving 6D object control through precise actions like pushing and flipping.

**Manipulation with Legged Robot:** Legged robots are increasingly recognized for their potential in manipulation tasks. Leveraging research that integrates leg and arm mechanisms [40] and biomimetic designs, these robots enhance movement and object transportation [41]. Existing methods predominantly use mounted arms [1, 2, 42] or the robot's body [5, 6] for basic tasks, with few employing legs. In this context, some works have extended quadruped capabilities by attaching small grippers at the end-effectors of their legs to facilitate prehensile skills [43]. Others have utilized the legs for broader, large-scale manipulations, such as door pushing and basket lifting [3, 44, 45, 9] or rotating large yoga balls [46] but sacrificing mobility. In contrast, our approach emphasizes non-prehensile manipulation, using the quadruped's legs to directly control objects with precise actions like pushing and flipping smaller items, achieving 6D object control without relying on additional appendages and maintaining full mobility.

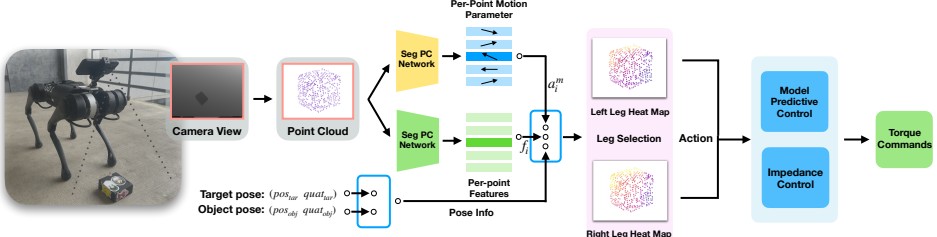

Figure 2: **Visual Manipulation with Legs.** Our system operates in stages: 1) A depth camera captures the object's point cloud. 2) The point cloud and target pose are processed by our network. 3) The manipulation leg is chosen based on the highest Q-value. 4) Pre-contact, contact, and action details are sent to the low-level control system. 5) The control system uses impedance control to direct the selected leg, while a Model Predictive Controller maintains balance, sending torques to the robot.

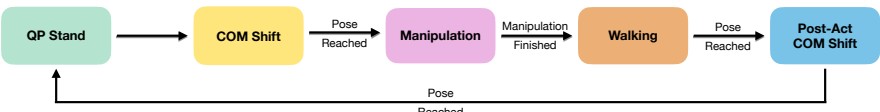

Figure 3: **Control FSM.** Our Finite State Machine (FSM) transition design follows a closed-loop approach, allowing for the repeated execution of manipulation actions through such a design.

Our system employs visual cues for detailed manipulation, an area not extensively explored in legged robotics. By combining advanced perception technologies with agile quadruped legs, we enhance the robots' precision and adaptability for complex tasks.

## 3 Visual Manipulation with Legs

We define our task as Object Pose Alignment; further details can be found in appendix A. An overview of the proposed system is summarized in Figure 2. Our proposed system includes two modules: 1) a learned visual policy that takes a point cloud observation of the object and outputs an action consisting of a contact location and a vector of motion parameters. 2) a model-based loco-manipulation controller that uses the front leg of the robot to interact with the object while maintaining the robot's balance and moving the robot for long-horizon object manipulation.

Our system integrates the visual policy and the model-based loco-manipulation controller to achieve object manipulation tasks. The model-based loco-manipulation controller operates within a framework that alternates between motion and manipulation states using a finite state machine structured around MPC. This approach requires multiple transitions between these states to accomplish a task. Every locomotion state is guided by the previous observation, setting a target position for the robot to approach. Upon reaching the target, the system transits into the manipulation state. Here, our learned visual policy computes and executes the necessary actions based on the current object and the goal object pose. Once the action is completed, the system reverts to the locomotion state to continue towards the next segment of the task, repeating this process until the task is completed.

### 3.1 Learning Visual Manipulation Policy

Our visual manipulation policy is trained with reinforcement learning using point cloud observations similars to [47]. The policy operates in an object-centric action space, selecting a contact location among the observed object points and a vector of motion parameters that define the post-contact movement [10]. Upon action selection, the robot moves its foot to the chosen contact location on the object. Once close enough, the robot executes the motion parameters, a 3D vector defining the push direction after contact. The range of motion parameters exceeds the robot's leg reach, allowing the policy to learn to apply varying force magnitudes for object manipulation.

The policy is trained with off-policy reinforcement learning algorithms based on TD3 [48]. We use a segmentation-style network architecture (e.g., PointNet++ [49]) for both the actor and the critic. Given a point cloud observation, the actor outputs per-point motion parameters (actor map), and the

critic outputs per-point Q-values (critic map). The policy output is determined by selecting the point with the highest Q-value from the critic map and the associated motion parameters.

### 3.1.1 Selecting from left or right legs

We propose using the critic values to select between the left or right legs during manipulation. The legs on quadruped robots usually have a restricted range of motion due to limited degrees of freedom and joint limits. Choosing between the front legs during manipulation can broaden the available motion possibilities. As our experiments will show, using the right leg for tasks on the right side of the robot is more efficient and results in a higher success rate.

We implement a policy capable of selecting the appropriate leg for tasks by modifying the structure of the actor map and critic map. Each point is associated with two motion parameters and two Q-values corresponding to the left and right legs. We select the point with the highest Q-value and its corresponding motion parameter and leg to complete the task. At testing time, a single point cloud is processed once, facilitating decision-making for both legs simultaneously.

### 3.2 Model-Based Loco-Manipulation Controller

Our loco-manipulation controller serves two purposes: 1) interact with the target object using the learned visual policy, and 2) move the robot for long-horizon object manipulation.

Built on an MPC controller [11] and [50], our controller divides the process into stages represented by a finite-state machine (FSM) in Figure 3: QP Stand, COM Shift, Manipulation, Walking, and Post-Act COM Shift. At each control step, the robot starts in QP Stand, where it stands with four legs. The visual policy computes the action from the object point cloud. The robot then shifts to COM Shift, lifting one leg to prepare for interaction. In Manipulation, the lifted leg moves to the pre-contact point and manipulates the object. MPC calculates torque for the stance legs, and impedance control is used for the swing leg. After manipulation, the swing leg returns to the ground in Post-Act COM Shift state, preparing the robot for walking. If the object is out of reach, the controller moves the robot closer.

**Model Predictive Control.** Our MPC, based on Bledt et al. [11] and Chen et al.[50], uses ground reaction force over a finite horizon $k$ to determine optimal control inputs and trajectory:

$$\min_{x,u} \sum_{i=0}^{k-1} ||x_{i+1} - x_{i+1,ref}||_{Q_i} + ||u_i||_{R_i} \text{subject to } x_{i+1} = A_i x_i + B_i u_i, \underline{c}_i \leq C_i u_i \leq \bar{c}_i, D_i u_i = 0,$$

Here, $x_i$ is the state, $u_i$ the control input, $Q_i$ and $R_i$ are weight matrices, $A_i$ and $B_i$ describe system dynamics, and $C_i$, $\underline{c}_i$, $\bar{c}_i$ set control constraints. $D_i$ identifies forces for feet not in contact. MPC commands include roll, pitch, yaw, Cartesian position, and target velocities. We primarily manipulate yaw (Z-axis) and provide linear velocity for X and Y directions. More details in Bledt et al. [11].

**Impedance Control.** Our impedance control adjusts mechanical impedance for optimal leg movement, enabling precise object interaction. Control torques ($\tau$) are computed as:

$$\tau = J^T \left( K_p(p_{\text{des}} - p_{\text{foot}}) + K_d(v_{\text{des}} - v_{\text{foot}}) \right),$$

where $p_{\text{des}}$ and $v_{\text{des}}$ are the desired foot position and velocity, $p_{\text{foot}}$ and $v_{\text{foot}}$ are the actual position and velocity, $K_p$ and $K_d$ are proportional and derivative gains, and $J$ maps foot force to joint torques.

This control algorithm achieves precise control over the robot's manipulating leg, enabling optimized interaction with the environment and task execution.

### 3.3 Point-cloud Registration Module

To perform accurate 6D object pose alignment, our system relies on the relative transformation between the target and current poses. While simulation provides accurate 6D poses, obtaining them in the real world is challenging, especially with egocentric visual observation.

We integrate RPM-Net [51], a learning-based point-cloud registration method, into our framework. RPM-Net processes source and target point clouds with per-point normals, outputting the transformation from source to target. We fine-tune the pre-trained RPM-Net on a subset of ModelNet40 [52]

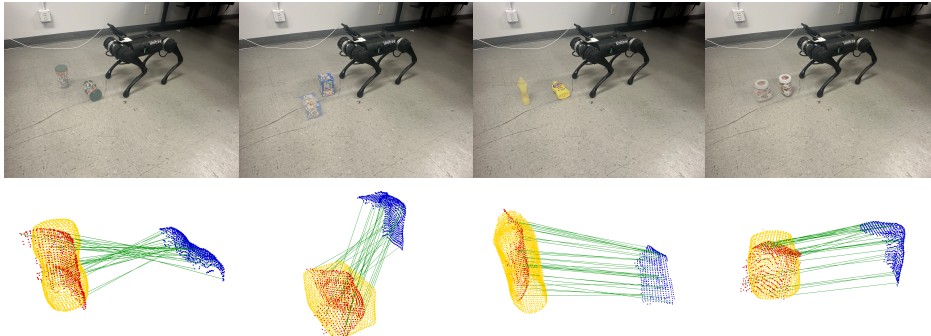

Figure 4: We employ RPM-Net for the registration process of a real robot, emphasizing successful registrations. In the illustration, the blue point cloud represents the source data captured by the camera, the yellow point cloud corresponds to the complete scan of the object, and the red point cloud shows the source data after transformation. Green lines indicate the flow vectors.

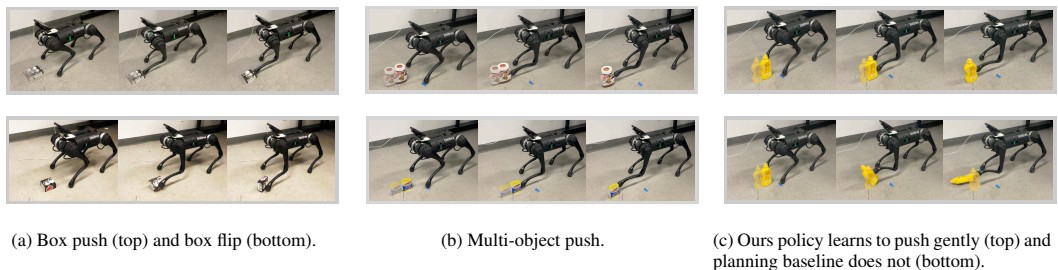

(a) Box push (top) and box flip (bottom).     (b) Multi-object push.     (c) Ours policy learns to push gently (top) and planning baseline does not (bottom).

Figure 5: We visualize the real robot trajectories. The semi-transparent overlies are the goal poses.

with domain-specific augmentations, including perturbed point-cloud normals and visual occlusions. Further fine-tuning on YCB [53] objects improves accuracy. At deployment, the source point cloud comes from the robot's camera, and the target point cloud is a object full scan set to the desired pose.

To enhance registration quality at inference, we augment the source point cloud with 6 random rotations, feed the augmented point clouds into RPM-Net, and use the best result for our visual manipulation policy. We evaluate registration quality using flow distance and Chamfer distance metrics. Flow distance favors minimal rotational adjustment, while Chamfer distance assesses average similarity between the registered object and target model. By ranking results according to both metrics and applying a preference weighting to Chamfer distance, we select the registration outcome with the lowest cumulative rank. Known rotations allow recovery of the registration result to the original pose. Point-cloud registration examples are shown in Figure 4.

## 4 Experiments

We evaluate our system and compare it with various baselines in simulation and the real world on object pose alignment and present quantitative and qualitative results in this section.

### 4.1 Experiment Setup

**Task.** We evaluate all methods on the following 5 tasks.

- **Box push (Fixed goal)**: Push a box forward (along x-axis in the robot frame) by 15cm.
- **Box push (Random goals)**: Push a box to a target position (in the robot frame) on the ground. The X and Y coordinates of the target position are uniformly sampled from $(10cm, 20cm)$ and $(-5cm, 5cm)$ respectively.
- **Box flip + push (Random goals)**: Flip a box left or right by 90 degrees and push to a target position sampled as in the *Box push (Random goals)* task.
- **Multi-object push (Fixed goal)**: Push an object forward (along x-axis in the robot frame) by 15cm.
- **Multi-object push (Random goals)**: Push an object to a target position sampled as in the *Box push (Random goals)* task.

For *Box push* and *Box flip + push (Random goals)* task, we randomize the length, width and height of the box within $(5cm, 10cm)$, $(5cm, 10cm)$ and $(4cm, 8cm)$ respectively. For *Multi-object push* tasks, we use 37 objects from Liu et al. [54], where we use 27 objects for training and 10 objects for evaluation. The object visualization is provided in the supplementary section. Sampled trajectories in the real-world experiment are visualized in Figure 5.

**Locomotion Strategy.** In the real deployment, our robot uses MPC to approach the object. MPC calculates foothold positions based on gait settings, allowing the robot to walk stably to the target. Initially, the target is placed in front of the robot without base movement. Later, the object's position is obtained through point-cloud registration. Commands for MPC are calculated using the robot's current position and the object's position, as described in Sec 3.2. In the simulation, we directly set the robot's position in front of the object with small variations, eliminating the need to walk over.

**Baselines.** We compare with the following baselines to verify the necessity of learning contact location and motion parameters:

- **Random Location Baseline**: Selects a contact location uniformly from the object point cloud and learns motion parameters for manipulation as our method does.
- **Flow Baseline**: Maintains contact location selection but replaces motion parameters with the flow between the object and goal point cloud at the selected point.
- **Planning Baseline**: For pushing tasks, strategically select the center point of an object's side as the contact point; for flipping tasks, select the midpoint of the edge in the direction of the top side flip. Tuned motion parameters ensure effective interaction with minimal unintended movements or rotations. These parameters are then applied to other objects.

**Training and Evaluation.** Our policy (except for leg-selection) and all baselines are trained in IsaacGym [55] for 50,000 steps, while the leg-selection policy requires 100,000 steps for optimal performance. Each policy is trained with three different seeds. The main evaluation metric is the success rate, defined as the mean flow between the object and goal being smaller than 3 cm. The mean flow measures the average distance between corresponding points in the current and target point clouds, considering both rotational and translational errors. By default, the evaluation uses the front left leg. Section 4.2 explores the advantages of selecting either the left or right leg.

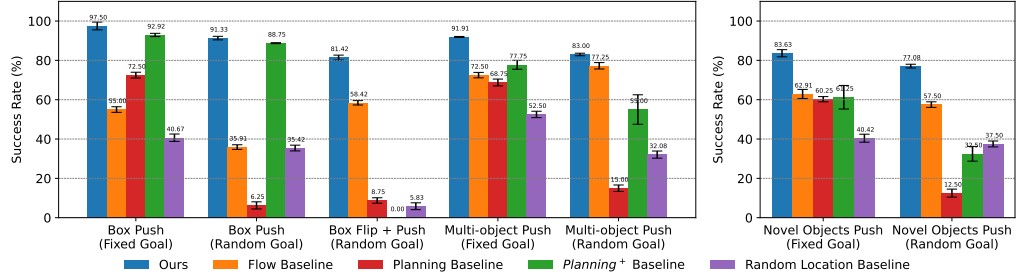

Figure 6: **Quantitative Results in Simulation and Generalization.** We evaluate the performance of our method against flow, planning, and random location baselines in various object manipulation challenges. The left plot shows results for training objects across different manipulation tasks. The right plot demonstrates generalization to novel objects in a pushing task. Our method consistently outperforms the baselines across all tasks.

## 4.2 Evaluation in Simulation

We first evaluate our method on single-box manipulation tasks, with results shown in Figure 6 (left). Our method achieves nearly 100% success on *Box push (Fixed goal)*, and over 80% success on the harder *Box push (Random goals)* and *Box flip + push (Random Goals)* tasks. Our method significantly outperforms all baselines on all three tasks.

Learning to select a contact point significantly outperforms random selection. The performance gap between the flow baseline and our method highlights the importance of learning motion parameters for accurate object manipulation, even with a simple box.

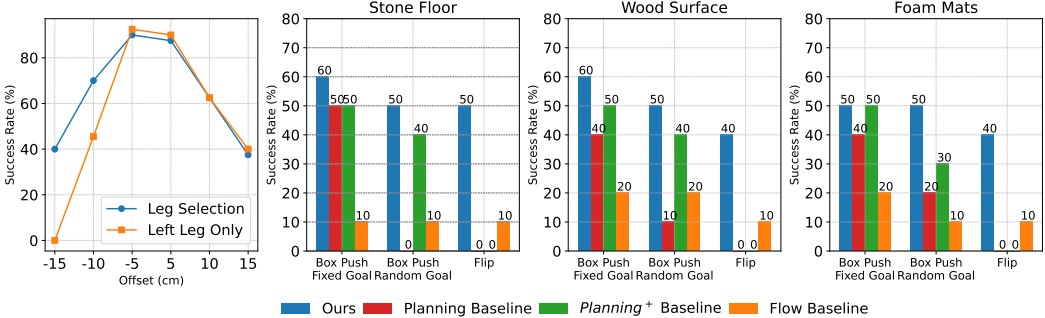

Figure 7: **Leg Selection and Real Experiment.** (Left) Success rates of single-leg control vs. our method on the *Box Flip + Push (Random Goal)* task with varying initial object positions. (Right) Average performance from ten real-world tests on various object manipulation tasks on three different type of surfaces.

When the robot needs to push the box forward, our visual manipulation policy outputs a vector pointing forward and slightly downward, enabling the robot to obtain larger friction and control the box's movement. The flow baseline's forward-only vector cannot effectively control the box, leading to poor performance.

When we expand our object set from a single box to diverse objects, our method still achieves over $80\%$ on both *Multi-object push (Fixed goal)* and *Multi-object push (Random goals)* tasks.

To further illustrate the superiority of our method, we compare the trajectory of our method and the flow baseline manipulating the same object in simulation in Figure 5c. When the policy manipulate a tall bottle, our method learned to use motion parameters that result in a gentle push. In contrast, without adjusting the motion parameters according to the object, the flow baseline flipped over the bottle and failed to push it forward.

Table 1: **Box Multi-Step.** Tasks fail if they exceed 20 steps or have a y-error over 20 cm. Data in the table shows the average from 5 real-world tests. We define the Y-error as the horizontal axis error, and it is measured with an April tag attached to the box and manually verified.

| Method | Steps | y-error (cm) |
|--------|-------|-------------|
| Ours | $12.8 \pm 1.89$ | $7.35 \pm 2.08$ |
| Flow | $>20$ | $>20$ |

**Generalization to unseen objects.** To evaluate the generalization capability of our proposed system across object shapes, we evaluated the policy trained with 27 objects on 10 novel objects and provided the results in Figure 6 (right). We found that our learned visual manipulation policy generalized well to novel objects without a significant performance drop, while the performance of other baselines dropped significantly.

**Ablation of leg selection.** As stated in Section 3.1.1, our model outputs actions for both legs, selecting the one with the highest Q-value.

We compare our method with a variant that only uses the left front leg for the *Box Flip + Push (Random Goal)* task. The object's location is randomized laterally in front of the quadruped, and the success rates for different object locations are shown in Figure 7 (left). When the object is close to the left front leg (offset $\geq -5cm$), both variants perform similarly. However, the left-leg-only variant fails when the object is farther away (offset $\leq -5cm$), while our method consistently outperforms it. This shows our leg selection strategy significantly enlarges the robot's operation space for manipulation tasks.

**Visualizations of the Policy Output** To better understand our system, we visualized the critic map in our learned visual manipulation policy, which scores each contact location on the object (Fig. 8). These Critic Maps capture goal-conditioned object affordances, providing insights into manipulation strategies. Figure 8a shows two scenarios involving pushing different objects, while Figure 8c focuses on flipping and pushing a box to the target pose.

We deploy our system in the real world for all 5 tasks using Unitree Go1 [56]. We quantitatively evaluated our method on *Box Push (Fixed Goal)*, *Box Push (Random Goal)*, and *Box Flip + Push (Random Goal)*. Each task had 10 trials, and the average success rates are provided in Figure 7 (right).

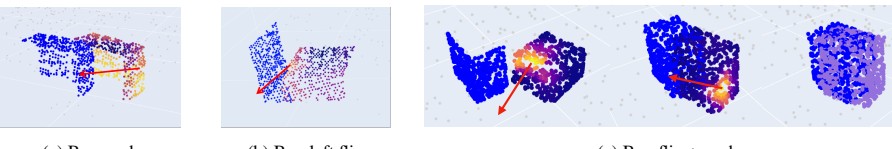

(a) Box push      (b) Box left flip      (c) Box flip + push

Figure 8: Blue points represent the goal point cloud. The color map shows the observed object point cloud (purple points in Picture 8c are the final object's point cloud), with lighter colors indicating higher critic map scores. Red arrows show motion parameters at selected contact locations. The policy selects different contact locations based on object geometry and goals.

### 4.3 Evaluation in the Real World

We compared our method with flow and planning baselines in real-world scenarios, omitting the random location baseline due to poor simulation performance. Our method achieved up to 60% success rate, while other baselines barely succeeded. Despite a smaller simulation gap, the flow baseline performed significantly worse in real-world tests. We hypothesize it is more sensitive to environment parameters like friction, object mass, and the sim2real gap in low-level control.

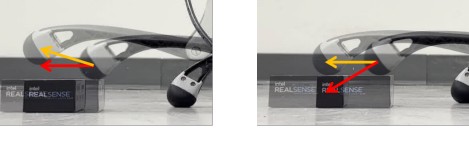

(a) Flow baseline      (b) Ours

Figure 9: **Low-level Control Error.** Red arrows denote commanded movements, yellow arrows represent actual movements.

Figure 9 illustrates this gap with a scenario where the robot needs to push a box forward by 10 cm. The flow baseline outputs a 10 cm forward vector, but control errors cause the actual foot trajectory to move upward, losing contact with the object. In contrast, our method compensates for this error by maintaining contact and completing the task.

**Moving the object to a distant goal.** Our system combines locomotion and manipulation, allowing the robot to move objects beyond its leg's reach. In an experiment, the robot pushed a box 1 meter forward, maintaining minimal lateral deviation. The manipulation policy, trained with close goals, repeatedly commands the robot to push the object 10 cm forward until it reaches the 1-meter target. After each action, the robot updates its observation using RPM-Net and adjusts its position for the next action with MPC. We evaluate performance with two metrics: steps to reach 1 meter and y-axis error at the final destination. Results in Table 1 show our system achieves higher accuracy and lower y-axis error compared to the baseline.

## 5 Conclusion

We propose a system that allows quadruped robots to manipulate objects with their legs using visual input. Our RL policy leverages point cloud observations for object interaction, with actions implemented via impedance control and MPC. We evaluate the system in both simulation and real-world, demonstrating significant advancements in legged manipulation skills not seen in previous work.

**Limitation:** While our system explores using a quadruped leg as a manipulator, it has limitations. For example, using an iPhone's lidar introduces accuracy issues in point cloud observations due to distortion from reflective surfaces, affecting the registration results of RPM-nets. Additionally, our reliance on MPC during training, though beneficial for sim-to-real transfer, faces challenges due to the lack of parallel processing for the Quadratic Programming (QP) problem, which slows down training. The current state estimator, relying on the robot's internal IMU, is susceptible to translation and rotation errors, which can result in inaccuracies in motion parameters and contact location during execution. Additionally, the front-mounted camera's inability to capture the object's side point cloud further restricts the robot's action space.

**Future work:** To further improve our approach, we could accelerate the QP solution by reducing the number of state variables through motion parameterization as quintic splines, as proposed in [57]. Additionally, incorporating friction estimation [58] or utilizing a friction dataset as a prior [59] could enhance the system's ability to generalize manipulation tasks to more complex terrains, addressing some of the challenges we identified.

**Acknowledgments**

We sincerely thank Yiyu Chen for the invaluable support in our MPC work and assistance with the real robot experiments, and Bowen Jiang for his guidance and help during the training phase. This project was supported in part by the Amazon Research Award, the Intel Rising Star Faculty Award, and gifts from Qualcomm, Covariant, and Meta.

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

## Appendix

## A    Task: Object Pose Alignment

In this work, we focus on object pose alignment. The objective is to align an object's 6D pose with a given target pose, represented as a transformation relative to the current object pose. Using an observed object point cloud and the target transformation, the quadruped robot interacts with the object using its front legs to achieve alignment. By default, the left leg is used, except in the leg selection experiment (Section 4.2)

## B    Environment Settings

We conducted our training within IsaacGym [60], where certain parameters of our simulator are detailed in Table 2.

Table 2: Environment Settings

| Parameters | Values |
| --- | --- |
| Environment numbers | 80 |
| Simulation dt | 0.001 |
| Object point cloud size | 400 |
| Background point cloud size | 400 |

## C    Reward Definition

We first define goal flow first: Suppose that point $x^i$ in the initial point cloud corresponds to point $x'^i$ in the goal point cloud. Then the goal flow is given by $\Delta x^i = x'^i - x^i$ . Then at each timestep, we define reward$r_t$ at timestep $t$ as the negative of the average goal flow as below, where $\|.\|$ denotes the L2 distance and N is the number of points in object point cloud:

$$r_t = -\frac{1}{N}\sum_{i=1}^{N}\|\Delta x_i\|,$$

## D    Camera Settings

During our training process, it was necessary to utilize the simulator's depth camera; the parameters for the depth camera, as utilized in our experiments, are presented in Table 3.

Table 3: Camera Settings

| Parameters | Values |
| --- | --- |
| Camera width | 80 |
| Camera height | 721 |
| Camera horizontal_fov | 71.36 |
| Camera offset | (0.24, 0, 0.14) |
| Camera rotation | Around Y-axis by 65° |
| Simulation dt | 0.001 |
| Object point cloud size | 400 |
| Background point cloud size | 400 |

## E    MPC Parameters

For robot control, we employed Model Predictive Control (MPC), with the control parameters for the MPC outlined in Table 4. The "MPC_weights" item corresponds to the weight matrices $Q_i$ of the

MPC mentioned earlier. The weights in the table, from left to right, are for *roll*, *pitch*, *yaw*, *position (XYZ-Axis)*, *angular_velocity (XYZ-Axis)*, *velocity (XYZ-Axis)*, and *gravity_place_holder*.

Table 4: MPC Parameters

| Parameters | Values |
|---|---|
| MPC_weights | [0.25, 0.25, 10, 50, 50, 50, 0, 0, 0.3, 0.2, 0.2, 0.2, 0] |
| MPC update frequency | 1 time / sim_dt |
| MPC force update frequency | 1 time / 50 sim_dt |
| MPC Cartesian K_p | diag(450, 450, 450) |
| MPC Cartesian K_d | diag(10, 10, 10) |

With the MPC illustrated earlier, The dynamics can be succinctly expressed as:

$$\mathbf{X} = \mathbf{A}_{qp}\mathbf{x}_0 + \mathbf{B}_{qp}\mathbf{U}, \tag{1}$$

where $\mathbf{X} \in \mathbb{R}^{13k}$ represents the vector encompassing all state variables over the prediction horizon, and $\mathbf{U} \in \mathbb{R}^{3nk}$ denotes the vector of all control inputs within the same period.

The objective function, aiming to minimize the weighted least-squares discrepancy from a reference trajectory alongside the weighted magnitude of forces, is formulated as:

$$J(\mathbf{U}) = \|\mathbf{A}_{qp}\mathbf{x}_0 + \mathbf{B}_{qp}\mathbf{U} - \mathbf{x}_{\text{ref}}\|_{\mathbf{L}} + \|\mathbf{U}\|_{\mathbf{K}}, \tag{2}$$

where $\mathbf{L} \in \mathbb{R}^{13k \times 13k}$ and $\mathbf{K} \in \mathbb{R}^{3nk \times 3nk}$ are diagonal matrices containing weights for state deviations and force magnitudes, respectively. Here, $\mathbf{U}$ and $\mathbf{X}$ represent the control input and state vectors over the prediction horizon. We assign equal weighting to forces with $\mathbf{K} = \alpha\mathbf{1}_{3nk}$.

More details can be found in Bledt et al. [11].

## F  RPM-Net Training Details

In terms of our employment of RPM-Net [51], we initiated training with the first 20 categories of the ModelNet40 [52] dataset employing a pre-trained model provided by the RPM-Net authors. This model was trained with cropping augmentation to perform registration on the incomplete point cloud. We opted not to use the normals from ModelNet, instead performing normal estimation on all objects using Open3D [61]. We then performed further augmentation by introducing visual occlusion through hidden point removal algorithm, increasing the extent of noise and cropping and better aligning with deployment. Training continued until convergence, which was achieved after roughly 1300 epochs. Subsequently, we fine-tuned the model previously trained in the described process on the YCB [53] dataset and our own scanned objects, continuing until convergence was once again attained. The fine-tuning process achieved roughly 8 degrees in rotation mean absolute error (MAE) and 0.05 in translation MAE. The visual cropping was conducted from random viewpoints, making it challenging to guarantee the remaining percentage of the object, though it was at most 70%.

This approach leverages the capabilities of RPM-Net for robust feature extraction and point cloud processing, enabling us to perform object pose estimation with the egocentric camera. The application of occlusion, noise augmentation, and fine-tuning on diverse datasets underscores our method's effectiveness in improving model generalization and accuracy in real-world scenarios.

# G  Training Hyperparameters

Table 5: Hyperparameters

| Hyperparameters | Values |
|---|---|
| Batch_size | 64 |
| Gradient_step | 160 |
| Discount factor ($\gamma$) | 0.99 |
| Location policy temperature(left_leg policy) ($\beta$) | 0.1 |
| Location policy temperature(leg_selection policy) ($\beta$) | 0.05 |
| Initial_timesteps | 10000 |
| Learning rate | 0.0001 |
| Max_episode_steps | 7 |
| Reward_scale | 1 |
| Critic clamping | [-20, 0] |
| Critic update freq per env step | 2 |
| Actor update freq per env step | 0.5 |
| Target update freq per env step | 0.5 |
| MLP size | [128, 128, 128] |

Our approach extends the Twin Delayed DDPG (TD3) algorithm [48], utilizing the framework provided by Stable-Baselines3 [62] as a foundation. For the segmentation-style network, we adopt PointNet++ segmentation architectures for both actor and critic networks, leveraging PyTorch Geometric's implementation. Details on hyperparameters can be found in Table 5. Both the actor and the critic networks are configured with identical architectures and learning rates.

# H  Training and Testing Object Sets

Below, we showcase the object we used in multi-object training. The blue objects are used for training, and the coral objects are used for evaluation.

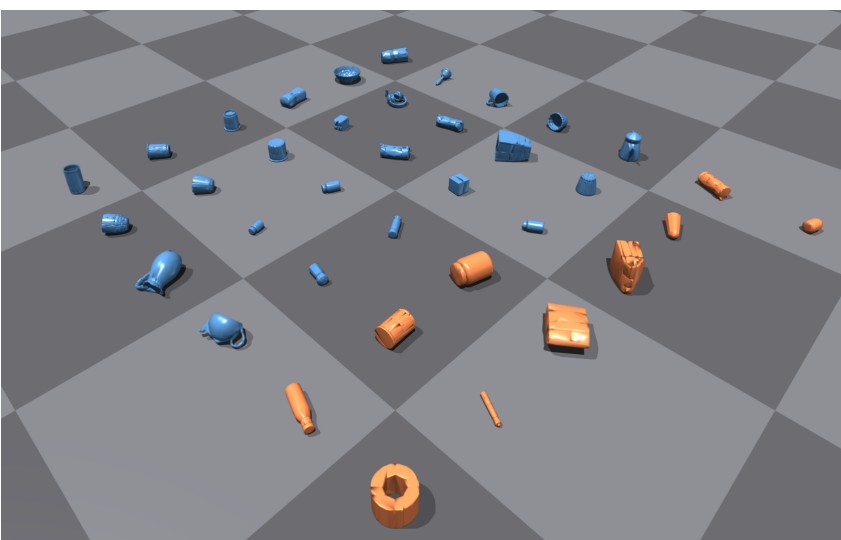

Figure 10: We showcase the objects we used in multi-object tasks; the objects in blue are used for training, and objects in orange are used for testing.

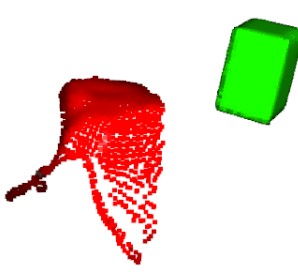

Figure 11: Registration Error: The red point cloud on the right represents the captured object, while the green point cloud indicates the target object pose. The registration result mistakenly shows the object as flipped.

Figure 12: Noisy Point Cloud Observation: The red point cloud represents the object captured by the camera, which is highly distorted compared to the target point cloud in green. This distortion leads to poor registration results and inaccurate policy outputs.

# I   Failure Case Analysis

One issue we observe is that noisy point cloud observations in the real world often lead to unreasonable outputs from our policy, as shown in Fig.12. This presents challenges for both RPM-net and our policy. Additionally, in our real robot experiments, a common failure case is registration error, particularly when objects lack distinct features. As illustrated in Fig.11, the registration result can incorrectly suggest that the object should be flipped, even during simple planar motions such as pushing an object forward. This error causes the pushing policy, which is trained on correctly oriented objects, to produce unexpected behaviors and ultimately fail the task. Control errors also contribute to failures, particularly in tasks where precise contact point accuracy is critical, such as in flipping tasks. As shown in Fig.13, the intended contact point (marked by the red dot) differs from the actual contact location (marked by the blue dot) captured by the camera. This discrepancy between the planned and executed contact points can lead to ineffective manipulation, resulting in the task's failure. These control errors highlight the challenges of achieving high precision in real-world scenarios, where even small deviations can significantly impact performance.

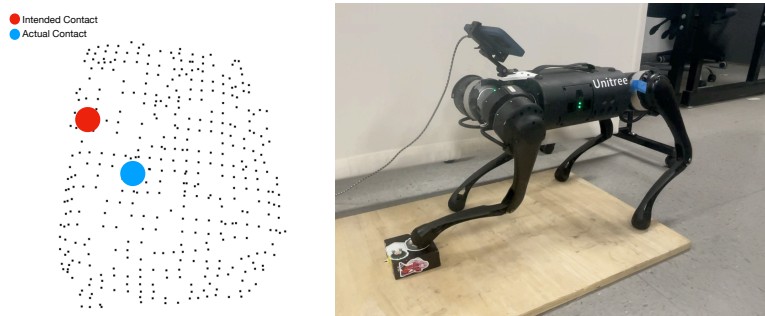

Figure 13: Control Error: The point cloud on the left is captured by the camera, with the red dot representing the intended contact point and the blue dot indicating the actual contact location.

