# OpenReview forum: "Visual Manipulation with Legs"
_robot-learning.org/CoRL/2024/Conference — CoRL 2024_

### Official Review · Reviewer_xy5P · 2024-07-10
**An intuitive path forwards for legged manipulation**

**Originality:** 3
**Technical Quality:** 3
**Clarity Of Presentation:** 4
**Potential Impact:** 3
**Recommendation:** 3
**Confidence:** 4

**Review:**

Overall, the paper is presented clearly and concisely and represents an interesting but intuitive path forwards for legged manipulation. The authors have done a good job presenting the proposed method, however there are concerns regarding the thoroughness of the literature review as well as issues regarding figure quality and the thoroughness of the experimental validation.

Major Comments:
1) Figures:

a) Figures 1, 2, 4, 5, 6, 9, and 10 are all low-quality and pixelated, making inferring information from them challenging. In particular, Figures 4,5, and 6 are illegible. Saving the original images in svg or pdf format and importing these should fix this issue.

b) Figure 10 in the Appendix is unlabeled and does not contain a caption.

c) The point clouds in Figure 2 are very blurry and unrecognizable as point clouds.

2) While the experiments show improved results for the proposed method when compared to the baselines, the range of object goal translations is limited in the y-direction. I would be interested in seeing how the proposed method behaves when the robot is required to push an object sideways, for instance in a circle around the robot, or pull the object closer to itself.

3) In Section 5.3, the second paragraph suddenly ends.

4) In Section 5.3, the authors indicate that “[...] our method compensates for this error by maintaining contact and completing the task.” While I agree with the authors that the proposed method outperforms the flow-based baseline, I disagree that this is because the policy has learned to compensate for controller error. From the authors descriptions, this control error does not appear to be present in the IsaacGym simulation environment, and so the policy has learned nothing about this error. I would appreciate a more in-depth examination of why the proposed method outperforms the baselines.

5) The author’s breakdown of the literature in Section 2 is confusing. There is no contribution to the blind locomotion literature, and so this area should not be included. Furthermore, there is no discussion of the non-prehensile manipulation literature, an area of research which this paper claims to advance.

6) Section 6, regarding the limitations of the work, is not thorough. There appear to be several possible limitations which the authors do not address, neither in this section nor the experimental sections. First, the proposed approach assumes the object to be manipulated is light enough to be pushed or flipped by the robot, which may not be the case in the real world. I would be interested to see what happens when such a failure case occurs. Second, it appears the impedance control for the leg is open-loop with respect to contact forces. If the robot does not make contact with the object, what happens?

Minor Comments:
1) It may be helpful to parameterize each of the state variables in the MPC optimization using a quintic spline (refer to Section III.A of [1]). This minimized the number of optimization parameters especially when there is a fine time discretization.

2) The authors may be interested in parallel QP software such as this library.

3) The authors may be interested in recent work regarding physical property (friction, mass, etc) estimation from vision [2][3]. While it is not necessary to integrate into this work, combining the proposed approach with these methods may be an interesting future research direction.

[1] Dario Bellicoso, Fabian Jenelten, Christian Gehring, and Marco Hutter. "Dynamic locomotion through online nonlinear motion optimization for quadrupedal robots." IEEE Robotics and Automation Letters 3, 2018.
[2] Parker Ewen, Adam Li, Yuxin Chen, Steven Hong, and Ram Vasudevan. "These maps are made for walking: Real-time terrain property estimation for mobile robots." IEEE Robotics and Automation Letters, 2022.
[3] Donghun Noh, Hyunwoo Nam, Min Sung Ahn, Hosik Chae, Sangjoon Lee, Kyle Gillespie, and Dennis Hong. "Surface material dataset for robotics applications (SMDRA): A dataset with friction coefficient and rgb-d for surface segmentation." International Conference on Pattern Recognition (ICPR), 2020.

**Quality Of The Limitations Section:**

1

**Questions For Rebuttal:**

I would like to see an addition experiment involving pushing an object through a non-linear path which requires the robot to push the object sideways. Furthermore, it would be interesting to see what happens if the object is unable to be pushed by the robot, for instance if the object was too heavy.

Additionally, the authors must present a more thorough literature review regarding non-prehensile manipulation to better ground their work in the this field.

Lastly, the authors should present a more thorough evaluation of the limitations of this work.

**Robotics Focus:**

4

**Summary Of Paper:**

This paper presents a learning-based approach for non-prehensile manipulation using the legs of a quadrupedal robot. The manipulation policy is trained in simulation and outputs motion vectors for the quadruped’s leg. This output is then tracked using an impedance controller while MPC is used to maintain balance and center-of-mass control. The approach is validated across experiments involving pushing and flipping different objects.

**Summary Of Recommendation:**

This paper presents an intuitive research direction for legged manipulation which I have not seen in the literature, but several issues need to be addressed within the paper

---

### Official Review · Reviewer_VjUa · 2024-07-24
**Systems-type paper lacking novelty for method and task**

**Originality:** 2
**Technical Quality:** 4
**Clarity Of Presentation:** 4
**Potential Impact:** 2
**Recommendation:** 2
**Confidence:** 4

**Review:**

Overall I think the work is a bit minor. The novelty is pretty limited and is more of a system putting together different previously researched techniques. The task is not particularly novel either, with plenty of other works doing non-prehensile manipulation with quadruped legs, pushing boxes, pushing buttons, opening doors, etc. What seems to be the main point of the work, doing learned visual manipulation, is just the previously published HACMan method applied to the author’s specific quadruped platform. The only main differences seems to be an improvement in the point cloud registration method, using RPM-Net as opposed to an out of the box global registration method with ICP. As such the contribution of this work is mostly just serving as another example of visual (non-AprilTag) manipulation for quadrupeds, of which there are still few.


## Strengths:

Though minor, the leg selection module is a good logical extension of the HACMan per point Q-value setup. Having a “per-point-per-leg” Q-value makes sense.

The system does seem to perform pretty well. When it does succeed, visually the objects end up quite close to the final goal pose. Generalization to different objects and even different ground surfaces are shown in the video results as well (though no accuracy evaluations are given for different ground surfaces).

The paper is presented well and is very understandable. I had zero technical questions while reading. Exact details of the method are a bit light, but they are heavily referenced and hyperparameters are given in the appendix, so it is still clear exactly what is being used and in sufficient detail. Enough explanation is given that someone could recreate this work. The technical details of the methods seem to be all correct to my understanding.

A lot of hardware experiments are done, which is very appreciated. Multiple hardware trials for each task are also done.

## Limitations:

The novelty in this work seems quite limited, outside of these specific techniques being applied to this specific task. All of the individual techniques used, such as the learned visual manipulation policy, the MPC system, and object pose estimation and alignment are all pretty much just the referenced works (your references [10], [16], and [54] respectively) directly applied to your task. The leg selection for the manipulation policy is just a minor extension of the per-point Q-value and highest Q-value selection that the HACMan method does.
There is also not much novelty to the task, as there are prior works doing learned controllers for quadruped non-prehensile manipulation, see [1-5] (these should also probably be part of your related works section). On the visual side of things [6] also uses PointNet++ for manipulation tasks (they use it for grasping rather than non-prehensile manipulation, but is a very similar example of using PointNet++ to go from point cloud observation to grasping/contact points).
This main novelty is in incorporating vision in to these manipulation tasks, and is more of a systems paper of applying HACMan to a quadruped rather than a fixed based manipulator (though once the robot is standing on 3 legs and balanced through the MPC controller it is basically liked a fixed base arm). The point cloud registration part is improved upon, but the main idea of training and system setup is the same as HACMan.

There is a pretty large drop in performance between the simulation results and hardware results. What are the causes of failures on hardware? Does the robot not execute the motion parameters correctly, does it not contact the object at the right location, is the object pose transformation estimation not accurate enough (i.e. the RPM-Net point registration has errors), etc.? You can test some of these in sim as well, how does the system perform when you add noise in the point cloud observation and/or object pose estimate (speaking of which, is the policy trained with any of these noise/randomizations?) I think it’s good to report not just the performance and accuracy of your system, but to also try and show why the failures happen if possible. Right now only point cloud observation noise is mentioned in the limitations. These are also things that you can add to there as well. You can also probably add in/look at the limitations mentioned in HACMan since you use the same method with no changes here as well.

The Flow baseline is a good baseline to compare against to isolate the effect that the learned contact selection has, but I think having a baseline that shows the effect of the learned motion parameters is good/necessary to have as well. Basically my question is “what is more important for this task, selecting the contact location well or outputting good motion parameters”? The “Random Location Baseline” is part of answering this, but I would like to see something like the planning baseline with the learned motion parameters. If we have a decent heuristic for what contact location to use, and then just learn the motion parameters, how well does it do?


References:

[1] He, Zhengmao, et al. "Learning Visual Quadrupedal Loco-Manipulation from Demonstrations." arXiv preprint arXiv:2403.20328 (2024).

[2] Shi, Fan et al. “Circus ANYmal: A Quadruped Learning Dexterous Manipulation with Its Limbs.” 2021 IEEE International Conference on Robotics and Automation (ICRA) (2020): 2316-2323.

[3] Arm, Philip, et al. "Pedipulate: Enabling manipulation skills using a quadruped robot’s leg." 41st IEEE Conference on Robotics and Automation (ICRA 2024). 2024.

[4] X. Cheng, A. Kumar and D. Pathak, "Legs as Manipulator: Pushing Quadrupedal Agility Beyond Locomotion," 2023 IEEE International Conference on Robotics and Automation (ICRA), London, United Kingdom, 2023, pp. 5106-5112, doi: 10.1109/ICRA48891.2023.10161470.

[5] Lin, Changyi, et al. "LocoMan: Advancing Versatile Quadrupedal Dexterity with Lightweight Loco-Manipulators." arXiv preprint arXiv:2403.18197 (2024).

[6] P. Ni, W. Zhang, X. Zhu and Q. Cao, "PointNet++ Grasping: Learning An End-to-end Spatial Grasp Generation Algorithm from Sparse Point Clouds," 2020 IEEE International Conference on Robotics and Automation (ICRA), Paris, France, 2020, pp. 3619-3625, doi: 10.1109/ICRA40945.2020.9196740.

## Formatting comments:

Can you increase the text size in Fig. 2. Right now it is too small to easily read at a reasonable zoom level.

Would it be possible to increase the image resolution of Figure 4 and 5? Right now it’s hard to see the individual points and get an idea of what is going on in the point transformation and the critic map. Same for Figure 6, it’s almost useless to have because it’s so small and low resolution that can hardly tell how the object is moving. This kind of goes for all images that aren’t generated plots actually. Even figure 2 is pretty low res, why does it look like that when the same figure in your video looks much better? There might be some pdf compression issues going on, perhaps any image that isn’t an SVG is just getting really really compressed. Something to double check.

On line 241 the sentence “Our method achieved up to 60” ends abruptly, seems like there is some missing text here? Judging from Figure 8 I’m guessing you meant something like “Our method achieved up to 60% success rate”?

If space allows could you move Figure 5 to be later in the text so it’s closer to the where you reference it in line 227? It was weird seeing early and not having the context for what it means/is supposed to so, and then not reading anything about it until 2 pages later.

**Quality Of The Limitations Section:**

2

**Questions For Rebuttal:**

In section 4.2 what is “Post-Act COM Shift”? Why is this step needed? The explanation of each fo the FSM states in lines 115-122 seems to end after the “Walking” state.

The visual policy takes in a point cloud observation of just the object correct? So it is already segmented out from the raw point cloud observation? What is used for the point cloud segmentation?

For the Planning Baseline, what are the “tuned motion parameters” used? Is this something the heuristic used in the Flow baseline, where it’s the flow between the object and goal location? Or is just some fixed push direction? Is this why performance drops so much when the goal is randomized vs. when it is fixed?

Along a similar line, you mention that your visual manipulation policy outputs a vector pushing the object forward and slightly downward, which performs much better than the Flow baseline’s forward-only vector. If you add some downward direction to the motion parameter to the Flow baseline (and the Planning baseline as well) how well does it perform? This seems like a very reasonable heuristic to use. Overall I am surprised that for what seems like (at least to me) a relatively simple task of pushing an object to a goal, a simple heuristic of selecting the object center point and then pushing forward and slightly down would perform so bad.

How does the MPC controller decide how much closer to move the robot to the object in the long horizon cases? Looking at the videos it seems like the robot could move much closer to the object and thus push the object farther in each manipulation step, moving it to the goal much quicker. Is there just some set target distance of how far the robot should be in relation to the current pose of the object?

**Robotics Focus:**

4

**Summary Of Paper:**

This paper presents a system for doing non-prehensile manipulation with the legs of a quadruped. The target task is pushing objects along the ground to a target pose using a single leg of the robot. This is done by using a MPC controller to balance the robot on 3 legs so it can move the 4th with an impedance controller to push the object. Employing the HACMan method, the manipulation leg is controlled by a learned visual policy, which takes in point cloud observations and outputs an object contact location along with motion parameters to define the pushing motion. To do pose alignment RPM-Net is used to do point cloud registration and to get the transformation between the current object pose and the target pose. Evaluations are done in simulation and on hardware, comparing against different heuristic baselines. The proposed method outperforms all baselines and can generalize to unseen objects on the real robot.

**Summary Of Recommendation:**

The work lacks a lot of novelty in both method and task and mostly serves an having another example of visual non-prehensile quadruped manipulation. I am not convinced enough that this is a large enough contribution for acceptance, but having more (especially hardware) examples in the field is not a bad thing when there are still relatively few. I would lean towards reject but can swing either way given the hardware experiments shown.

---

### Official Review · Reviewer_AEx8 · 2024-07-26
**Improving quadruped robot non-prehensile manipulation by sim-to-real high level visual policy and low-level model-based controller**

**Originality:** 3
**Technical Quality:** 3
**Clarity Of Presentation:** 3
**Potential Impact:** 3
**Recommendation:** 3
**Confidence:** 3

**Review:**

Strengths
- The system effectively combines visual 3D point clouds and control techniques to allow quadruped robots to manipulate objects with their legs. The high level policy outputs object-centric actions in terms of target object motion direction and desired contact location on the object. The low level controller is a model-based controller to transform the target motion on a single front leg (left or right) into torques.
- The approach demonstrates successful real-world application, showing the system's practical use with performance conparison over a few baselines shown in the paper.

Weaknesses
- In the evaluation comparison, the task seems to be limited in terms of the object diversity, goal definition, and object-environment interactions (frictions, etc.). The robot intends to either push objects to a location or flip objects on the ground.
- The learned policy seems to be not very autonomous with a few heuristically-defined behavior transitions among standing and walking. The task seems not affecting robot dynamics too much, which could reduce the impact of this work when considering applying it to legged robots.

**Quality Of The Limitations Section:**

2

**Questions For Rebuttal:**

Expand the range of tasks and objects to include more diverse and complex interactions, addressing limitations in object diversity, goal definition, and object-environment interactions (e.g., friction variations). This could involve tasks beyond pushing and flipping, like lifting or rotating objects or using legs to do side motions than just forward push.

**Robotics Focus:**

4

**Summary Of Paper:**

The paper shows a system that lets four-legged robots use their legs to move objects. The system has two parts: a visual policy trained to see and decide how to move objects, and a controller that moves the robot's legs and body. This allows the robot to push or flip objects to new places using either leg. Real-world settings show the system works better than a few baselines. The paper's key contributions are combining high level visual policy with low level model-based movement control and testing on different tasks.

**Summary Of Recommendation:**

The paper introduces a promising system for enabling quadruped robots to use their legs for object manipulation, combining visual data and control techniques. However, the current tasks are limited to pushing and flipping objects, and the system relies heavily on predefined behavior transitions, which reduces its autonomy. Expanding the task diversity and improving the system's adaptability to dynamic tasks would enhance its real-world applicability. Addressing these issues would strengthen the paper's contributions and make the system more robust and impactful.

---

### Author Rebuttal · Authors · 2024-08-14

# Rebuttal ZIP File

- Revised paper with modified part highlighted with blue.
- Rebuttal videos with additional experiments.

Please also see the comments to individual reviewers for details.

---

### Decision · Program_Chairs · 2024-09-04

**Decision:**

Accept

**Comment:**

The reviewers point out several strengths and weaknesses of the paper. While the method has good results and this indeed a challenging task, more baselines could be considered and the method uses some heuristics which take away from the generality of the approach. Please see reviewer comments for more details.

-----Post-rebuttal comments----
The rebuttal addresses some of the key issues brought up by reviewers and the paper appears to be above the bar for acceptance.